# Mirtazapine Reduces Adipocyte Hypertrophy and Increases Glucose Transporter Expression in Obese Mice

**DOI:** 10.3390/ani10081423

**Published:** 2020-08-14

**Authors:** Ching-Feng Wu, Po-Hsun Hou, Frank Chiahung Mao, Yao-Chi Su, Ching-Yang Wu, Wei-Cheng Yang, Chen-Si Lin, Hsiao-Pei Tsai, Huei-Jyuan Liao, Geng-Ruei Chang

**Affiliations:** 1Division of Thoracic and Cardiovascular Surgery, Department of Surgery, Chang Gung Memorial Hospital, Chang Gung University, Linkou, 5 Fuxing Street, Guishan District, Taoyuan 33305, Taiwan; maple.bt88@gmail.com (C.-F.W.); wu.chingyang@gmail.com (C.-Y.W.); 2Department of Psychiatry, Taichung Veterans General Hospital, 4 Section, 1650 Taiwan Boulevard, Taichung 40705, Taiwan; peterhopo2@yahoo.com.tw; 3Faculty of Medicine, National Yang-Ming University, 2 Section, 155 Linong Street, Beitou District, Taipei 11221, Taiwan; 4Department of Veterinary Medicine, National Chung Hsing University, 250 Kuo Kuang Road, Taichung 40227, Taiwan; fcmao@nchu.edu.tw; 5Department of Veterinary Medicine, National Chiayi University, 580 Xinmin Road, Chiayi 60054, Taiwan; shu@mail.ncyu.edu.tw (Y.-C.S.); tsaibelle@mail.ncyu.edu.tw (H.-P.T.); pipi324615@gmail.com (H.-J.L.); 6Department of Veterinary Medicine, School of Veterinary Medicine, National Taiwan University, 4 Section. 1 Roosevelt Road, Taipei 10617, Taiwan; yangweicheng@ntu.edu.tw (W.-C.Y.); cslin100@ntu.edu.tw (C.-S.L.); 7College of Veterinary Medicine, Veterinary Teaching Hospital, National Chiayi University, 580 Xinmin Road, Chiayi 60054, Taiwan; 8Ph.D. Program of Agriculture Science, National Chiayi University, 300 Syuefu Road, Chiayi 60004, Taiwan

**Keywords:** adipocyte, blood glucose, insulin, mirtazapine, obesity

## Abstract

**Simple Summary:**

Mirtazapine, a tetracyclic antidepressant, acts through noradrenergic and specific serotonergic systems. Consequently, it was recently applied in major depressive disorder treatment. Moreover, because mirtazapine may have effective glucose control function, its mechanism of action warrants further investigation. In our study, we examined how mirtazapine affects metabolic parameters, insulin profiles, glucose metabolism, and obesity changes in high-fat diet-fed C57BL6/J mice. Our results indicated that compared with untreated mice, mirtazapine-treated obese mice had lower insulin levels, daily food efficiency, body weight, serum triglyceride levels, aspartate aminotransferase levels, liver and epididymal fat pad weight, and fatty acid regulation marker expression. Moreover, the blood glucose levels and area under the curve for glucose levels observed over a 120 min assessment period were lower in the treated mice, but the insulin sensitivity and glucose transporter 4 expression levels were higher in these mice. They also demonstrated a considerable decrease in fatty liver scores and mean fat cell size in the epididymal white adipose tissue, paralleling adenosine monophosphate (AMP)-activated protein kinase expression activation. In conclusion, mirtazapine administration may alleviate type 2 diabetes mellitus with hyperglycemia.

**Abstract:**

Metabolic syndrome is known to engender type 2 diabetes as well as some cardiac, cerebrovascular, and kidney diseases. Mirtazapine—an atypical second-generation antipsychotic drug with less severe side effects than atypical first-generation antipsychotics—may have positive effects on blood glucose levels and obesity. In our executed study, we treated male high-fat diet (HFD)-fed C57BL/6J mice with mirtazapine (10 mg/kg/day mirtazapine) for 4 weeks to understand its antiobesity effects. We noted these mice to exhibit lower insulin levels, daily food efficiency, body weight, serum triglyceride levels, aspartate aminotransferase levels, liver and epididymal fat pad weight, and fatty acid regulation marker expression when compared with their counterparts (i.e., HFD-fed control mice). Furthermore, we determined a considerable drop in fatty liver scores and mean fat cell size in the epididymal white adipose tissue in the treated mice, corresponding to AMP-activated protein kinase expression activation. Notably, the treated mice showed lower glucose tolerance and blood glucose levels, but higher glucose transporter 4 expression. Overall, the aforementioned findings signify that mirtazapine could reduce lipid accumulation and thus prevent HFD-induced increase in body weight. In conclusion, mirtazapine may be useful in body weight control and antihyperglycemia therapy.

## 1. Introduction

Mirtazapine, a tetracyclic antidepressant that acts through specific serotonergic and noradrenergic systems, has predominantly been evaluated for its alleviating effects on major depressive disorder (MDD) [1]. Although the exact mechanism underlying the therapeutic effects of mirtazapine remains unclear, mirtazapine is potentially as effective as other antidepressants and exhibits noradrenergic together with serotonergic effects; moreover, its onset of action is considered to be faster than that of selective serotonin reuptake inhibitors (SSRIs). The serotonergic effects may involve the enhancement of the effects mediated by serotonin (i.e., 5-hydroxytryptamine (5-HT)) receptors [2]. Furthermore, mirtazapine inhibits adrenergic receptors and heteroreceptors through the selective antagonization of adrenergic α2-autoreceptors and α2-heteroreceptors and through 5-HT_2_ and 5-HT_3_ receptor inhibition. Mirtazapine also inhibits presynaptic central α2-adrenergic receptors; this inhibitory effect intensifies noradrenaline and serotonin release [1,3], thereby enhancing norepinephrine and serotonin transmission. This mechanism is possibly responsible for the rapid onset of action of mirtazapine. Short-term (5–6-week-long) clinical trials have indicated that the drug is safe in overdose, with a considerably lower propensity for inducing seizure-related adverse events compared with tricyclic antidepressants [1]. However, in the long term, side effects were reported at the 16 week endpoint, with the most common being panic disorders and increased appetite and weight gain [4]. Furthermore, mirtazapine has been used as an appetite stimulant for healthy cats and cats with chronic kidney disease [5]. By contrast, mirtazapine and some SSRIs may reduce body weight through effects on neurotransmitters, neuronal ion channels, and other pathway components, including those on energy expenditure [6,7].

Obesity, a global health problem, is established as constituting a prominent risk factor for assorted metabolic-syndrome-related conditions, examples of which are nonalcoholic fatty liver disease (NAFLD), hyperlipidemia, type 2 diabetes mellitus (T2DM), insulin resistance (IR), and cardiovascular diseases [8]. A consumption–expenditure imbalance engenders a net increase in the body’s energy storage, potentially increasing body weight and leading to obesity [9]. However, some antipsychotic drugs, such as olanzapine, clozapine, and fluphenazine, inhibit glucose transport, potentially resulting in weight gain, impaired glucose tolerance, hyperlipidemia, and increased adiposity [10,11]. Nevertheless, in rodent and murine studies, administration of high aripiprazole, clozapine, and ziprasidone doses did not lead to weight gain [6,12]. In another study, applying conventional antipsychotics, which included pimozide, fluphenazine, and chlorpromazine, caused cellular glucose transporter (GLUT) 1 and GLUT3 levels to rise substantially [11]. Moreover, in rat cardiac myocytes, dopamine, 5-methoxytryptamine, tryptamine, and serotonin administration intensified the transport of glucose, and this effect was paralleled by amplified cell surface GLUT4 and GLUT1 expression [13]. Nevertheless, as indicated by these reports, the relevant literature includes inconsistent results; hence, researchers have yet to clarify the extent to which energy expenditure changes influence body weight variations, whether such changes vary according to the applied antipsychotic drugs, and whether the mentioned influence is mediated in glucose homeostasis control, which can be determined by evaluating the effects of antipsychotics on GLUT expression.

Most lines of research on the metabolic side effects of mirtazapine have focused on the central nervous system. The influence of mirtazapine on glucose homeostasis and fatty liver scores remain poorly understood, which are essential for the metabolic homeostasis of glucose and lipids. We executed the study reported herein with the objective of determining the metabolic effects of mirtazapine on obesity and hyperglycemia in obese animals. Accordingly, we administered mirtazapine treatment to high-fat diet (HFD)-fed mice to mimic the effects noted in obese animals and humans. Our results provide insight into the obesity and hyperglycemic mechanisms underlying the psychotropic and metabolic effects of long-term mirtazapine use in animals.

## 2. Materials and Methods

### 2.1. Animals, Dietary Induction of Obesity, and Mirtazapine Treatment

We procured four-week-old male C57BL/6J mice from Education Research Resource, National Laboratory Animal Center (Taiwan). We housed each mouse separately in a microisolation cage on HEPA filter-ventilated racks (Rungshin IVC Systems, Taichung, Taiwan); the cages had temperature and humidity settings of 22 (±1) °C and 55% (±5%), respectively, as well as a 12:12 h light–dark cycle. In addition, the mice were afforded unrestricted access to food as well as water. We fed the mice an HFD (#592Z, 20.4% protein along with modified Laboratory with 35.5% lard (4.5 kcal/g metabolizable energy); PMI Nutrition, MO, USA) continually for 10 weeks—longer than the duration generally used in the literature (i.e., 4 weeks)—to induce obesity [14]. Each group was then divided into two subgroups, one of which was treated with 10 mg/kg oral mirtazapine (Sigma-Aldrich, MO, USA) or saline (control) by daily gavage for the 28 days last days of the HFD-feeding duration (35.12 ± 0.51 g for mirtazapine-treated mice vs. 34.86 ± 0.27 g for control-vehicle mice, *p* > 0.05). The administered mirtazapine dosage was based on the literature regarding mirtazapine’s use as a candidate for long-term treatment in behavior, neuroscience, and oncology studies conducted on mice [15,16,17].

When the study was completed, we sacrificed the mice through anesthetic overdose, harvesting their specific tissues as well as blood serum to be used in our subsequently executed analyses. The executed analyses entailed evaluating the influence exerted by mirtazapine on fatty liver scores, blood glucose, insulin expression, body weight, adipocyte content, biochemical changes, endocrine profiles, and food intake.

The Taiwan government-recommended guidelines from Guide for the Care and Use of Laboratory Animals were followed. Moreover, National Chiayi University’s Institutional Animal Care and Use Committee inspected and authorized our study protocol (approval No.: 109019).

### 2.2. Insulin Level, Food Intake, and Body Weight Measurement

We executed weekly measurements of food intake as well as body weight throughout the study period. To conduct our food intake measurement, we weighed the leftover food within each cage dispenser in addition to that spilled on the floor. Moreover, after harvesting the tissues and blood of the mice, we measured serum insulin levels by using a mouse insulin enzyme-linked immunosorbent assay (ELISA) kit (#INSKR020; Crystal Chem Inc., Downers Grove, IL, USA).

### 2.3. Serum Triglyceride and Aspartate Aminotransferase Level Measurement

Serum triglyceride and aspartate aminotransferase (AST) levels were measured from the collected blood samples by using an automated chemistry analyzer (Catalyst One Chemistry Analyzer, IDEXX Laboratories, Westbrook, ME, USA), commercial kits, and the manufacturer-recommended methodology, all with a coefficient of variation within and between analysis runs of <2%.

### 2.4. Intraperitoneal Glucose Tolerance Test

After 21 days of saline or mirtazapine treatment, we performed an intraperitoneal (i.p.) glucose tolerance test (IPGTT) in the obese mice that were fasted overnight with free access to water; the dose applied in this test was 1 g of glucose/kg body weight and is applicable for evaluating anti-diabetes mellitus (DM) activities of several agents in animal models of obesity and DM [18,19]. We measured glucose levels in blood at the following time points after i.p. glucose injection: 0, 30, 60, 90, and 120 min. We acquired the blood employed in this measurement from the tail vein by employing a One Touch glucose meter (LifeScan, Malpitas, CA, USA). Over a period spanning from 0 to 120 min after glucose injection, we executed glucose tolerance assessments on the basis of area under the curve (AUC) values.

### 2.5. IR and Insulin Sensitivity Index

Fasting blood glucose levels are employed for estimating the functions of IR and insulin sensitivity (IS) [19,20,21] as the homeostasis model assessment (HOMA)-IR index and IS index (ISI) through HOMA (a method validated against clamp measurements), respectively. The HOMA-IR index can be derived herein as follows [20]:HOMA-IR = [fasting insulin (mU/L) × fasting glucose (mmol/L)]/22.5
Moreover, the ISI can be derived herein as follows:ISI = (1/[fasting insulin (mU/L) × fasting glucose (mmol/L)]) × 1000

### 2.6. Morphometric and Histological Analyses of Tissues

We measured liver and epididymal fat pad weights and present them herein as a percentage of the total body weight. We also executed hematoxylin and eosin (H&E) staining for the purpose of visualizing fat infiltration inside the liver, and we scored liver surface infiltration by fat as follows: 0, no visible fat, 0; 1, <5%; 2, 5–25%; 3, 25–50%; and 4, >50% [8,22].

We obtained numerous epididymal adipose tissue sections and systematically analyzed the adipocyte size in the H&E-stained sections, which were observed under a Moticam 2300 system (Motic Instruments, Richmond, BC, Canada), a high-resolution digital microscope. We analyzed ≥10 fields/slide (i.e., ~100 adipocytes) for each sample [19] and determined adipocyte size distribution by using the software program Motic Images Plus (version 2.0). Adipocyte size correlation between the control and treated mice was also evaluated.

### 2.7. Western Blotting

Immediately after the mice were sacrificed, the gastrocnemius muscles and liver were excised, coarsely minced, and homogenized. Next, we executed Western blot analysis by applying the procedures outlined in a previously executed study [22]. We employed anti-GLUT4, anti-fatty acid synthase (FASN), anti-phospho-AMP-activated protein kinase (AMPK) (Thr 172), anti-actin, and anti-AMPK antibodies procured from Cell Signaling Technology (Beverly, MA, USA) for the analysis. Subsequently, we detected immunoreactive signals by using enhanced chemiluminescence reagents (Thermo Scientific, Rockford, MA, USA) and then exposing the blot membranes to radiographic films. Thereafter, we quantified protein expression and phosphorylation by using National Institute of Health’s Scion Image (Scion, Frederick, MD, USA).

### 2.8. Statistical Analysis

We present all data herein as means ± standard errors of the mean (SEMs). We used the *t*-test to analyze the between-group differences and considered *p* < 0.05 to indicate significance. Moreover, we evaluated the significance of contingency data by using Fisher’s exact test.

## 3. Results

### 3.1. Mirtazapine Affects Morphometric Parameters and Food Intake and Efficiency

Our preliminary studies demonstrated no significant antiobesity effects in the mirtazapine-treated standard diet (SD)-fed mice (Appendix A). However, after 4 weeks of mirtazapine treatment, HFD-fed mice exhibited elevated levels of morphometric parameters compared with the control mice (Figure 1). We noted the mouse body weight and weekly body weight gain in the treated group to be reduced by 14% (*p* < 0.05; Figure 1A) and 57% (*p* < 0.001; Figure 1B), respectively, compared with the control group. By contrast, the treated mice demonstrated significantly higher weekly food intake (*p* < 0.05; Figure 1C), but significantly lower daily food efficiency (*p* < 0.01; Figure 1D). Therefore, the relatively low body weight identified in the treated mice may not have been due to reduced food intake.

### 3.2. Mirtazapine Reduces Liver and Fat Pad Weights

Next, the association of weight differences with liver or adiposity alterations was assessed. After 4 weeks of mirtazapine treatment, we executed a body composition comparison between the treated and control mice; the comparison results revealed the two groups to differ significantly in epididymal white adipose tissue (EWAT) and liver weights (Figure 2). The treated group was noted to exhibit a 50% and 41% reduction in liver weight (Figure 2A) and EWAT weight (Figure 2B), respectively (both *p* < 0.001). Moreover, the treated group was observed to exhibit a 15% and 42% reduction in body-weight-normalized liver weight and EWAT weight (both *p* < 0.001), respectively.

### 3.3. Mirtazapine Reduces Serum Triglyceride and AST Levels

We noted a 31% reduction in blood triglyceride levels—constituting a metabolic syndrome indicator—in the treated mice relative to the control mice (*p* < 0.001; Figure 3A). Furthermore, we identified a 67% reduction in serum AST levels, constituting a hepatic function marker, in the treated mice (*p* < 0.001; Figure 3B).

### 3.4. Mirtazapine Reduces Liver Fat Accumulation and Adipocyte Size

Our H&E staining-based morphometric analysis of tissue sections obtained from the mirtazapine-treated mice as well as control mice revealed considerable reductions in liver fat levels and EWAT adipocyte sizes in the treated mice when compared with the control mice (Figure 4A), indicating that mirtazapine prevents fat accumulation in the liver and thereby possibly reduces fat pad hypertrophy.

Our analysis of changes in fatty liver scores (Figure 4B) and EWAT adipocyte sizes (Figure 4C) indicated significant between-group differences. Specifically, the fatty liver scores derived in the treated group were noted to be lower than those derived in the control mice by nearly threefold (*p* < 0.001). Moreover, the mean EWAT adipocyte size was determined to exhibit a 39% reduction in the treated mice (*p* < 0.01), corresponding to our earlier observation of lower fat pad weights in this group. That is, we observed that the treated group exhibited higher proportions of EWAT adipocytes with diameters of 0–40 and 40–80 μm (Table 1), but lower proportions of EWAT adipocytes with diameters of 80–120 and >120 μm. Thus, although the HFD tended to increase the EWAT adipocyte size considerably, mirtazapine reversed this increase.

### 3.5. Mirtazapine Alleviates Glucose Intolerance and Reduces Insulin Level

The IPGTT conducted to compare the tolerance levels of the treated and control groups revealed significantly lower fasting blood glucose levels, along with alleviation of glucose intolerance, in the treated mice (*p* < 0.001; Figure 5A). Moreover, mirtazapine treatment engendered a significant reduction in fasting blood glucose levels as observed at the postinjection time points of 30 (*p* < 0.05), 60 (*p* < 0.001), 90 (*p* < 0.01), and 120 (*p* < 0.001) min. In addition, at our postinjection time point of 120 min, we noted a 4% and 16% increase in such glucose levels in the treated and control mice, respectively, relative to their baseline levels (*p* < 0.001). We also determined that during the 120 min assessment period, the AUC for glucose levels in the treated group decreased significantly by 1.3-fold relative to that observed in the control group (*p* < 0.01; Figure 5B). We set our glucose intolerance criterion as a blood glucose level of >7 mmol/L at 120 min after injection; accordingly, we found that many treated mice exhibited glucose homeostasis with glucose intolerance alleviation (*p* < 0.001, Figure 5C). Nevertheless, we determined the serum insulin levels to be lower in the treated mice (*p* < 0.05; Figure 5D). Therefore, in our HFD-fed obese mice, mirtazapine treatment alleviated DM symptoms, including increased fasting blood glucose levels and increased glucose intolerance, as well as hyperinsulinemia.

### 3.6. Mirtazapine Enhances IS by Altering GLUT4 Expression, Reducing FASN Expression, and Lipogenesis

We assessed IR and IS in the mirtazapine-treated mice by using the HOMA-IR index and ISI, respectively [23]. Our comparison of the two groups revealed that the HOMA-IR index and ISI in the treated mice decreased by 2.7-fold and increased by 2.6-fold, respectively (both *p* < 0.05; Figure 6A,B, respectively). We identified the enhancement in glucose transport after mirtazapine treatment by evaluating the effects of mirtazapine on GLUT expression. Compared with the control mice, the treated mice demonstrated a 2.2-fold increase in muscle GLUT4 expression (*p* < 0.001; Figure 6C).

Fatty acid synthase (FASN) is a key enzyme in triglyceride synthesis and lipid homeostasis [10]. According to the result of Western blotting of the liver tissue, FASN expression in the treated mice was reduced by 5.3-fold relative to that in the control mice (*p* < 0.001). In addition, liver-specific adenosine monophosphate-activated protein kinase (AMPK) activation reduced lipogenesis in vivo and was key in the integration of metabolic pathways to address the energy demand [15]. This study also found that AMPK phosphorylation was enhanced by nearly 2-fold in the treated mice (*p* < 0.001).

## 4. Discussion

We executed the present study to probe the effects of mirtazapine on obesity development in HFD-fed C57BL/6J mice. The derived results reveal that mice treated with mirtazapine for 28 days had relatively low obesity rates, accompanied by lowering of visceral fat as well as development of glucose homeostasis and hypoinsulinemia. The effect of mirtazapine remained substantial and persisted over time. In other words, our results confirm that mirtazapine can retard body weight gain and reduce food intake, in addition to reducing glucose intolerance, serum insulin levels, daily food efficiency, adipocyte size, liver and epididymal fat pad weights, and fatty liver scores. Although body weight variations typically exhibit an association with food intake, our mirtazapine-treated obese hyperglycemic mice exhibited hyperphagia with considerably improved IR and IS. After 28 days of mirtazapine treatment, compared with control mice, the treated mice demonstrated significantly enhanced insulin signaling, corresponding to increased glucose homeostasis.

Over the 4 weeks of HFD feeding in mice, the HFD contributed to obesity development, whereas mirtazapine promoted body, liver, and EWAT weight reductions. Weight loss is a result of reductions in the fat pad mass caused by reduced adipocyte differentiation (delayed generation of adipocytes from precursor cells) or adipocyte hypertrophy (fat storage–induced reduction in adipocyte size) [8,21]. Thus, in our mirtazapine-treated mice, body fat loss may have resulted from reductions in the mean fatty liver score and adipocyte size. Furthermore, mirtazapine could reduce the number of large adipocytes, possibly because it promotes fat combustion, a finding consistent with reports in the literature that mirtazapine could reduce or inactivate the expression of adipogenic marker genes, including *ERK1/2*, *TGF-β*, and *SREBP* [24,25,26], which influence lipogenesis, adipogenesis, and HFD-induced adipocyte hypertrophy, respectively [8,27,28].

We next probed lipolytic activity and fatty acid oxidation enhancement by analyzing serum glycerol levels, adipose lipolytic/lipogenic protein levels, and fatty acid catabolic pathways in the treated mice. We found that the treated obese mice demonstrated increases in serum glycerol levels linked to lipolysis [29] (Appendix A). We also evaluated the effects of hormone-sensitive lipase (HSL) expression on lipolysis, fatty acid oxidation [30,31], and patatin-like phospholipid domain containing protein 3 (PNPLA3) expression in lipogenesis [32] (Appendix A). We found that increased HSL expression and decreased PNPLA3 expression in the EWAT of the treated mice considerably increased lipolysis and reduced lipogenesis, respectively. Furthermore, in our HFD-fed mice, mirtazapine treatment was found to inhibit FASN—a protein that plays a role in adipocyte differentiation and fat accumulation—expression [33]. Taken together, these results indicate mirtazapine can prevent obesity engendered by excess caloric intake; thus, mirtazapine may become beneficial for the identification of pivotal energy balance regulators considering its influence on lipolysis and lipogenesis.

In the literature, mirtazapine is suggested to increase appetite and thus contribute to body weight gain by increasing leptin resistance [34,35], a finding consistent with our results that indicate increased food intake in our treated mice. Nevertheless, our mirtazapine-treated mice gained body weight less rapidly than their control counterparts; this implies that mirtazapine raised lipid oxidation as well as energy combustion, consequently decreasing the size of adipocytes and engendering body weight reduction. The increase in energy consumption may account for the body fat loss. Fibroblast growth factor (FGF) 21 achieves energy metabolism regulation through AMPK-SIRT1-PGC-1α pathway activation [36]. These observations are consistent with findings regarding mirtazapine-induced increase of energy expenditure, reduction of lipid levels, and alleviation of hepatosteatosis and glycemia, all caused by increased FGF-21 levels (Appendix A); in both murine and nonhuman primate models of DM and obesity, FGF-21 is a key factor regulating the amelioration of glucose and lipid parameters [37]. Thus, we propose that the weight gain reduction was significantly enhanced in the mirtazapine-treated HFD-fed mice compared with the controls because mirtazapine induced an increase in the energy expenditure.

Several studies, however, have reported results contrasting our results. For example, mirtazapine was found to increase body weight gain in human clinical trials [38,39] and in pancreatic tumor-bearing mice [40]; moreover, no weight gain was noted in clozapine-treated mice or olanzapine-treated rats, but weight loss was observed in rats treated with high doses of antipsychotic drugs [12]. Here, the possible reason for the incongruity between the rodent and human data is the side effects of antipsychotics in animals, including sedation and muscle stiffness, all of which reduce activity and alter metabolism and thus interact with the influence exerted by the antipsychotics on food intake, satiety, and metabolism, particularly in short-term studies [41]. Nevertheless, animal studies have presented valuable information concerning the putative mechanisms underlying weight gain or weight reduction engendered by antipsychotics; however, results reported by such research have differed across handling and housing conditions, strains, drug administration methods, and species. Moreover, in the current study, a low dose (2 mg/kg) of mirtazapine had a slightly significant effect on the trend of weekly body weight gain in our HFD-fed mice (*p* < 0.1); however, this trend did not differ significantly when compared with the trend of body weight (Appendix A). Thus, additional studies focusing on body weight changes in murine or rodent models of depression treated with mirtazapine are warranted.

A previously executed study revealed that mirtazapine therapy—titrated to 45 mg at bedtime for approximately 8 weeks for MDD and obsessive-compulsive disorder treatment—may have led to increased serum triglyceride levels, consequently contributing to acute pancreatitis [42]. Another study on healthy subjects increased the daily dose of mirtazapine from 15 mg provided at the beginning to 30 mg in the second week. The results indicated that increased serum triglyceride levels were positively correlated with changes in weight over 4 weeks of treatment [43]. Therefore, although mirtazapine is associated with an increased appetite, it does not promote weight gain. Our findings demonstrate that continual mirtazapine use at 10 mg/kg/day for 4 weeks could reduce serum triglyceride levels, despite the higher food intake, as indicated by significant differences in body weight gain in our treated group relative to the control group. We further determined that reducing FASN expression could reduce triglycerides levels, alleviate hepatosteatosis, and improve IR [10,44,45]. Moreover, a reduction in circulating triglyceride levels could contribute to a reduction in hepatic lipid accumulation, which is the most important factor involved in the alleviation of NAFLD [46]. Here, the mirtazapine-treated HFD-fed mice exhibited lower fatty liver scores than the untreated mice. This effect may be attributed to the mirtazapine-mediated increase in the AMPK activity; this can be a viable therapeutic strategy for NAFLD alleviation in preclinical animal models. These results support those of other studies [47,48] and can be explained by the hepatic activation of AMPK, which protects against hepatosteatosis, dyslipidemia, and accelerated atherosclerosis in obese and insulin-resistant mice. Furthermore, our results revealed that mirtazapine treatment could improve the serum AST levels, a hepatic inflammation index. These results support the theory that inhibiting fatty liver formation activates AMPK levels and regulates nutrient metabolism. AMPK has anti-inflammatory activity in adipose tissue, and the mechanisms underlying this activity involve the suppression of multiple proinflammatory pathways, including interleukin (IL)-1 receptor-associated kinase-4 phosphorylation, in adipocytes [49]. In addition, the positive effects of mirtazapine in adipose tissues involve the AMPK-induced inhibition of multiple proinflammatory signaling pathways. Therefore, this mirtazapine-induced pathway regulation may provide new therapeutic targets for the common sequelae of obesity.

In a previous study on the L6 rat myoblast cell line, serotonin increased glucose uptake [50]. Moreover, in another study, serotonin treatment improved glucose tolerance in obese mice through a mechanism that does not increase GLUT levels in the plasma membrane [51]. Our findings are corroborated by the aforementioned results, particularly with regard to large reductions in blood glucose levels in response to increasing serum serotonin levels in the mirtazapine-treated HFD-fed mice (Appendix A). Here, we employed the IPGTT to evaluate the effects of mirtazapine on hyperglycemia in the HFD-fed mice and found mirtazapine treatment to have led to a considerable improvement in glucose tolerance, as illustrated by the reduced HOMA-IR index and the 120 min AUC for plasma glucose levels. A possible explanation for this result is the mirtazapine-induced increase in GLUT4 (a muscle GLUT protein) expression, phosphatidylinositol 3,4,5-phosphate levels, and protein kinase B activity, all leading to improvements in insulin signaling [50]. However, during pregnancy, serotonin may regulate glucose-stimulated insulin secretion from pancreatic β cells, according to the literature [51]. Moreover, SSRI-treated patients with MDD may demonstrate significant increases in insulin levels [52]. These findings contradict our finding that mirtazapine administration reduces serum insulin levels, despite higher serotonin levels that parallel hypoglycemia. All the aforementioned discrepancies may be due to the differences in the body weights of the study subjects or to the potential association of obesity with hyperinsulinemia or hyperglycemia [8,9,19]. Nevertheless, serotonin resistance as a disorder in major depression may lead to T2DM development. Further research focusing on the effects of mirtazapine on changes in tissue response to serotonin signaling is warranted.

Our results also indicate that the mirtazapine-treated mice demonstrated increases in ISI, a major pathophysiological indicator of insulin action and IR [19]. Increased adipose tissue weight and hepatosteatosis are possible risk factors for IR [53]. By contrast, in our study, the mirtazapine-treated HFD-fed mice exhibited decreases in fat pad weights and fatty liver scores, but the controls exhibited increases in these factors. AMPK activation was also noted in the mirtazapine-treated mice, a result that is consistent with a previous study’s report of glucose production inhibition in the liver after mirtazapine treatment [54]. Furthermore, although mirtazapine treatment led to glucose homeostasis, the treated mice exhibited a reduction in insulin levels. A partial explanation for this improvement in glucose tolerance is increased IS due to liver AMPK phosphorylation; moreover, the changes in the insulin dynamics are attributable to reductions in insulin secretion as well as clearance and in insulin-degrading enzyme expression in the liver [55]. Moreover, here, the increases in the insulin levels did not correspond to enhancements in the ISI. We also noted that mirtazapine administration possibly leads to improvements in insulin signaling and glucose homeostasis by increasing GLUT4 expression and serum FGF-21 levels, and it may serve as a metabolic regulator by inducing hepatic changes in glucose flux and improved IS [56]. This may be caused by reductions in hyperglycemia, as demonstrated by the alleviation of glucose intolerance in an IPGTT. Studies have found that treatment with mirtazapine and mianserin (a mirtazapine analog) in rats with type 1 DM can reduce blood glucose levels [57,58]. These findings suggest that mirtazapine affects IS by attenuating long-term HFD consumption-induced hypoglycemia because increased insulin signaling reverses the exacerbated metabolic state to ensure glucose homeostasis maintenance.

Mirtazapine, a noradrenergic and specific serotonergic compound, is currently approved as an effective psychotropic and antidepressant drug for long-term therapy in several countries worldwide, including the United States and some European countries. Mirtazapine may also be used to treat not only anxiety disorders and insomnia, but also vomiting and nausea. Because MDD treatment in clinical trials primarily requires long-term mirtazapine exposure [59], the current study has crucial clinical implications. Our findings regarding long-term mirtazapine use indicate mirtazapine’s potential to enhance glucose homeostasis and thus reduce glucose intolerance and IR risk, potentially followed by alleviation of hyperglycemia. Along with previous findings [57,58,60], these results indicate that mirtazapine treatment, particularly long-term treatment, can have beneficial effects on glucose homeostasis, such as reducing glucose levels and improving IS, in animals with DM. However, other clinical studies have reported that mirtazapine-treated animals experienced weight gain [42,61]. Nevertheless, most of these studies have used small samples, without inclusion of a placebo group; this limits their conclusions regarding mirtazapine’s effects on glucose tolerance. The present findings demonstrate the beneficial effects of mirtazapine treatment on the glucose homeostasis of obese and hyperglycemic mice. Therefore, when mirtazapine is administered clinically, caution must be exercised; moreover, additional studies with large samples are necessary to further clarify the possible applications of mirtazapine treatment in animals with obesity and DM.

Our results indicate that although an HFD could be administered to produce a mouse model of obesity along with hyperglycemia, continual mirtazapine administration in the resulting mice could retard body weight gain and reduce fat pad weight, serum triglyceride and AST levels, fatty liver scores, and adipocyte size, despite the increased food intake. This phenomenon may be associated with energy expenditure and AMPK activation for nutrient metabolism regulation. Moreover, administering mirtazapine to the obese mice alleviated hyperglycemia and enhanced glucose tolerance. Increased GLUT4 expression in skeletal muscles possibly contributed to the mentioned changes in glucose metabolism. These results reveal the presence of enhanced glucose homeostasis and IS. In addition, while the HFD sped up glucose homeostasis impairment in the mice, mirtazapine prevented the development of DM symptoms. Therefore, mirtazapine may be effective in MDD treatment, and long-term mirtazapine administration may be effective for DM and obesity control.

## 5. Conclusions

In sum, the current study provided novel evidence that continual mirtazapine administration in mice has considerable effects against obesity and hyperglycemia developed in response to HFD feeding. This phenotype was determined to be associated with the preserved metabolic homeostasis of decreased adipocyte hypertrophy and increased GLUT expression. Corresponding changes in food efficiency, weekly body weight gain, fat pad weight, serum triglyceride levels, body weight, serum AST levels, fatty liver scores, and adipocyte size were also noted. Moreover, the beneficial effects of mirtazapine in the regulation of adipose lipogenesis, as well as adipose lipolytic, adipose lipogenic, and fatty acid catabolism, may reduce susceptibility to obesity-related metabolic disturbances in humans and animals. Our study provides several lines of evidence demonstrating the mechanism underlying the mirtazapine-induced alleviation of hyperglycemia: that is, mirtazapine was noted to increase the expression of GLUT4, a component of insulin signaling, which was then associated with lowering of glucose intolerance and IR and enhancement of IS. Thus, the distinct antiobesity and lower hyperglycemia response elicited by mirtazapine in the HFD-fed mice indicate that the use of mirtazapine as a psychotropic agent may not lead to the development of the related metabolic abnormalities. Finally, future studies considering long-term mirtazapine use in animals and evaluating for cachexia with underweight and hypoglycemia are warranted.

## Figures and Tables

**Figure 1 animals-10-01423-f001:**
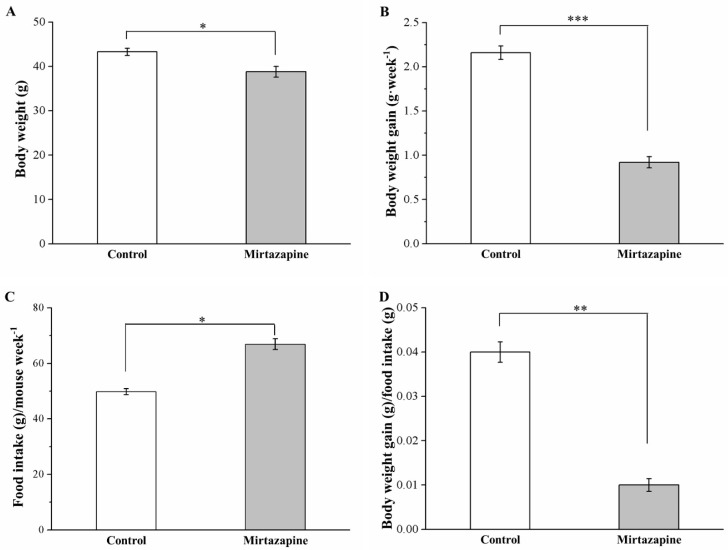
Changes in (**A**) body weight, (**B**) body weight gain, (**C**) food intake (per mouse per week), and (**D**) daily food efficiency in the control and 10 mg/kg/day mirtazapine-treated mice over the 28 treatment days. We present all data in this figure as mean ± SEMs (n = 10) for both groups. * *p* < 0.05, ** *p* < 0.01, *** *p* < 0.001.

**Figure 2 animals-10-01423-f002:**
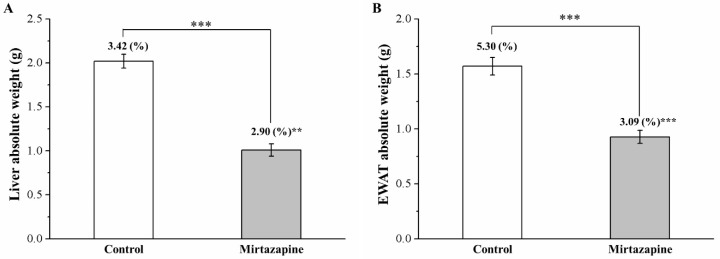
Changes in absolute weights (g) and body-weight-normalized weights (%) of (**A**) the liver and (**B**) the epididymal white adipose tissue (EWAT) in the control and 10 mg/kg/day mirtazapine-treated mice over the 28 treatment days. We present all data in this figure as mean ± SEMs (n = 10) for both groups. ** *p* < 0.01, *** *p* < 0.001.

**Figure 3 animals-10-01423-f003:**
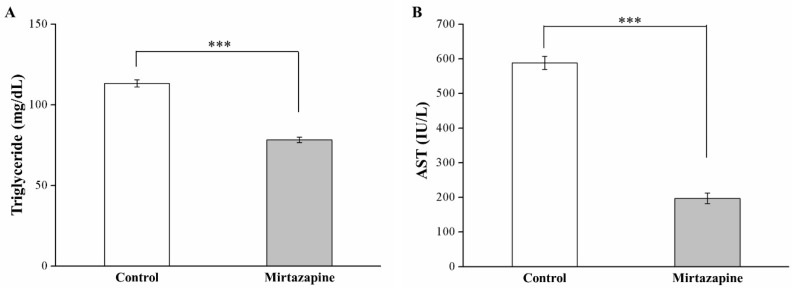
Changes in (**A**) plasma triglyceride and (**B**) aspartate aminotransferase (AST) levels in the control and 10 mg/kg/day mirtazapine-treated mice over the 28 treatment days. All data are presented as mean ± SEMs (n = 10) for both groups. *** *p* < 0.001.

**Figure 4 animals-10-01423-f004:**
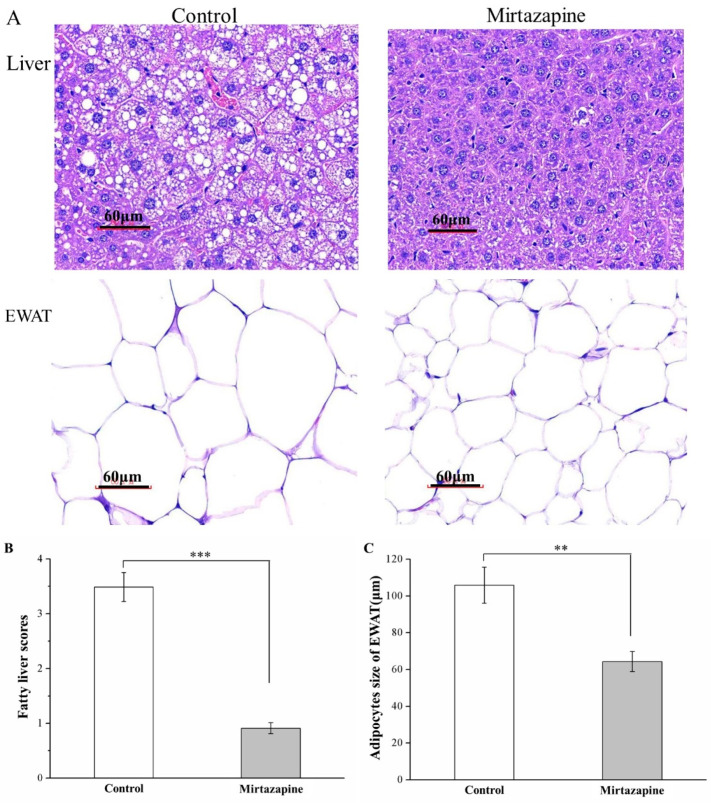
(**A**) Photomicrographs of H&E-stained liver and EWAT samples (magnification, 400×). Changes in (**B**) fatty liver scores and (**C**) adipocyte cellularity in the EWAT in the control and 10 mg/kg/day mirtazapine-treated mice over the 28 treatment days. We present all data in this figure as mean ± SEMs (n = 10) for both groups. ** *p* < 0.01, *** *p* < 0.001.

**Figure 5 animals-10-01423-f005:**
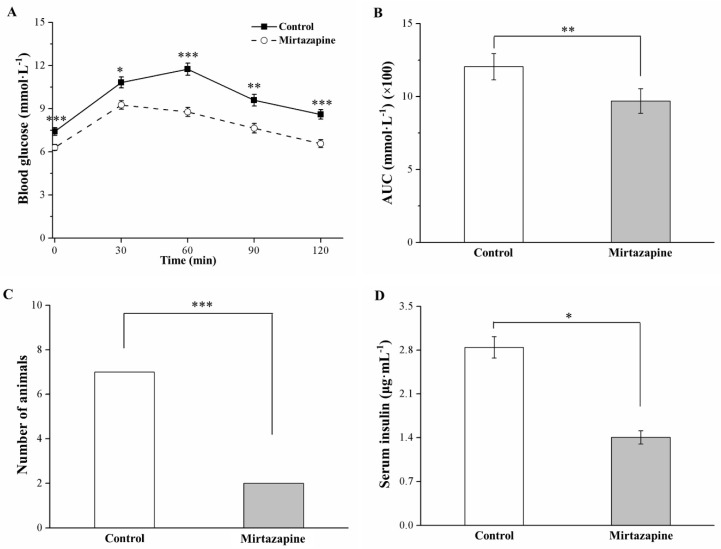
Changes in (**A**) intraperitoneal glucose tolerance test (IPGTT) (1 g of glucose/kg body weight), (**B**) area under the curve (AUC) over 120 min after glucose injection, (**C**) glucose intolerance criterion (Fisher’s exact test), and (**D**) serum insulin levels in the control and 10 mg/kg/day mirtazapine-treated mice over the 28 treatment days. We present all data in this figure mean ± SEMs (n = 10) for both groups. * *p* < 0.05, ** *p* < 0.01, *** *p* < 0.001.

**Figure 6 animals-10-01423-f006:**
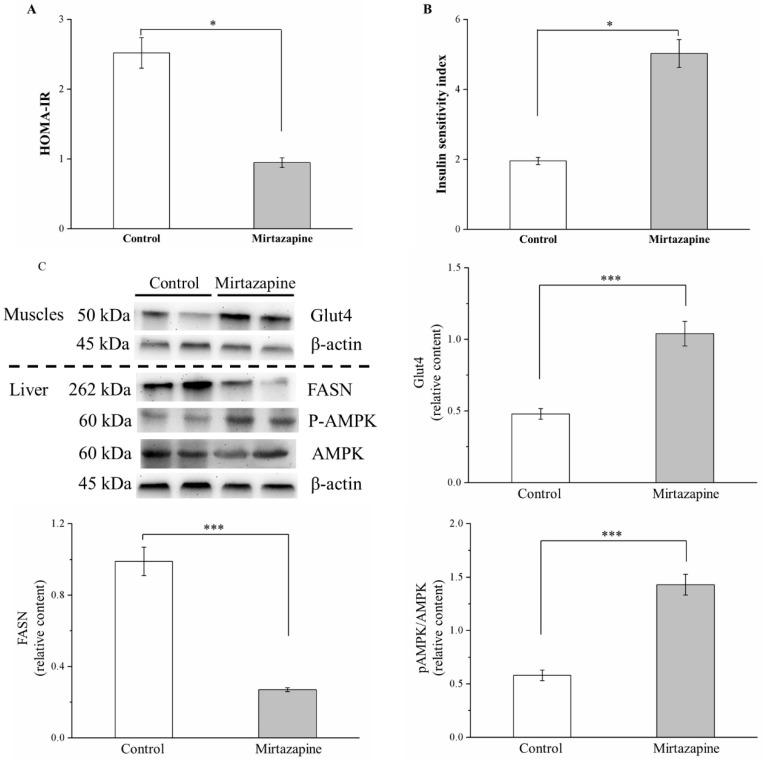
Changes in (**A**) homeostasis model assessment-insulin resistance (HOMA-IR) index, (**B**) insulin sensitivity index (ISI), and (**C**) GLUT4 expression in the gastrocnemius muscle and FASN and p-AMPK expression the liver in the control and 10 mg/kg/day mirtazapine-treated mice over the 28 treatment days. We present all data in this figure as mean ± SEMs (n = 10) for both groups. * *p* < 0.05, *** *p* < 0.001.

**Table 1 animals-10-01423-t001:** Changes in mirtazapine on fat cell size distribution of EWAT in the control and 10 mg/kg/day mirtazapine-treated high fat diet (HFD)-fed mice.

Variable	Control	Mirtazapine
0–40 μm (%)	3.92 ± 1.06	13.69 ± 1.09 **
40–80 μm (%)	10.17 ± 1.57	68.61 ± 0.65 ***
80–120 μm (%)	71.93 ± 1.75	17.69 ± 1.27 ***
>120 μm (%)	13.97 ± 1.72	0 ± 0 **

We present all data in this table as mean ± SEMs. ** *p* < 0.01, *** *p* < 0.001. n = 10 for both groups.

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
