# Peer review of "Mirtazapine Reduces Adipocyte Hypertrophy and Increases Glucose Transporter Expression in Obese Mice"

_animals, 2020, doi:10.3390/ani10081423_

Round 1

Reviewer 1 Report

This is section objective:

  1. determine the metabolic effects of mirtazapine on obesity and hyperglycemia in obese animals.

  1. The paragraph is to the section Methods. Insert is in Section Methods. The effects of oral mirtazapine on body weight, foodintake, blood glucose, adipocyte content, fatty liver scores, biochemical changes, endocrine profiles, and insulin signaling expression were evaluated.

Conclusion:

Only Conclusion:

Therefore, mirtazapine may be effective in the treatment of depressive disorders, and its long-term use may be effective in diabetes control or for animals with obesity

My suggesting:

Insert the paragraph in Section Discussion, in last paragraph.

This study demonstrated that an HFD can be used to generate an obesity mouse model with
hyperglycemia.

However, continuous administration of mirtazapine in obese HFD-fed mice led to a reduction in body weight gain, fat tissue weight, serum triglyceride levels, serum AST levels, fatty liver scores, and adipocyte size, despite the increase in food intake.

This phenomenon may be associated with energy expenditure and activation of AMPK for the regulation in nutrient metabolism.

Moreover, administering mirtazapine to obese mice alleviated hyperglycemia and enhanced glucose tolerance.

An increase in the expression of skeletal muscle GLUT4 may contribute to these alterations in glucose metabolism. These results revealed enhancements in glucose homeostasis and an increase in insulin sensitivity.

I have three question, please:

Why was this study done?

What did the researchers do and find?

What do these findings mean?

Thanks.

Reviewer 2 Report

In their manuscript entitled: “  Anti-obesity and hyperglycemic effect of mirtazapine  by decreasing adipocyte hypertrophy and increasing  glucose transporter expression in obese mice”,   Ching-Feng Wu et al.  present their results showing the potential benefits of using mirtazapine in obese mice.

MAJOR CONCERNS

Overall, this is a study with an interesting theme. However, the conclusions drawn are not efficiently supported by the data presented.

In particular, regarding the experimental groups under investigation one would also expect a “control group” in which the effects of mirtazapine would be evaluated under a standard diet. This is a fundamental flaw of the present study. Furthermore, authors should have included at least two different doses of the compound so as to enhance scientific soundness of their findings. Experimental groups used are not adequate for extrapolation of “safe” conclusions.

Authors state that “the dose of mirtazapine was based on the literature regarding mirtazapine’s use as a candidate for chronic treatment in behavior, neuroscience, and oncology studies in a mouse model”. However, the hypothesis under investigation in this particular study is completely different physiologically and therefore another dose could be required in order to test it. Unfortunately, the authors not only omitted using a control group but also chose to test the effect of one dose based on studies probing into completely different parameters.

As far as the experimental design is concerned, the authors note that: “At the termination of the experiment, animals were anesthetized for harvesting various tissues and blood serum for subsequent analysis”.  However, in these studies, animals are routinely deprived of food for at least 16 h before being sacrificed. Blood samples are subsequently collected, epididymal fat pads are removed and weighed. Segments of adipose tissue are fixed for histological inspection or frozen for analysis of lipolysis and HR-LPL activity and liver tissue is used for measurement of lipids. Biochemical analyses performed include: serum total cholesterol (TC), TG, low density lipoprotein cholesterol (LDL-C), high density lipoprotein cholesterol (HDLC), creatinine, uric acid, aspartate aminotransferase, alanine aminotransferase levels, as well as ketone body

(hydroxybutyrate) concentrations. Heparin-releasable lipoprotein lipase (HR-LPL) activity can also be assayed in adipose tissue.

In order to support the anti-obesity effect of mirtazapine and effectively decipher the mechanisms of action of the latter, authors should have shown reduction of de novo lipogenesis and triglyceride (TG) assembly, increased lipolysis and fatty acid oxidation.

A more detailed analysis of a number of mediators should have been performed and presented so as to better support the conclusions derived in this study. For example, when probing into anti-obesity mechanisms of action ane could look into:

Levels of phosphorylated acetyl-CoA carboxylase, levels of adiponectin receptor 1, CCAAT enhancer-binding protein α, CPT: carnitine palmitoyltransferase, FABP: fatty acid binding protein, LIPE: hormone sensitive lipase, LPL: lipoprotein lipase, PGC-1α: peroxisome proliferator-activated receptor gamma co-activator 1α and SIRT1: sirtuin 1, so as to demonstrate the mechanism of action of the compound of interest.

In its present form the MS is a descriptive study of a limited number of parameters including: body weight, food intake, serum insulin levels,  serum triglyceride and AST concentrations, glucose tolerance and fat infiltration in the liver. Of note, instead of choosing white adipose tissue or brown adipose tissue in parallel with skeletal muscle for their western blotting analysis, authors have exclusively studied expression levels of GLUT4, FASN and phospho-AMPK (Thr 172) in gastrocnemius muscle as they state in section 2.7, while in section 3.6 of the Results, the authors show that they have studied GLUT4 levels in muscle and FASN and phospho-AMPK in liver tissue….

The respective blots presented in Fig.6 are of low quality with no markers presented for verification of the molecular masses of the proteins. Why is there a double band detected in the blot for phospho-AMPK (Thr 172) ? There is also a significant difference between the signal detected from the duplicates in most cases that should also be explained.

Reviewer 3 Report

Wu et al have presented their work investigating the effects of Mirtazapine on several metabolic factors that influence body weight in high fat diet fed mice. They found that despite the animals treated with Mirtazapine eating more HFD, the animals weighed less and were more glucose tolerant than saline controls. The results presented were clear and the article is well written. 

Major comment:

Please double check all graphs are presented as mean +/- SEM not St Dev. If they are indeed SEM then I would doubt some of the significance presented in Figures 1A,C 2A, 3A. Especially if there is only N=10.

Minor comments:

Please add one sentence in the methods on why the dose of 1g/kg bw of glucose was chosen rather than the traditional 2g/kg?

Figure 1C - Please change the Y axis title to week instead of day. The text states "week" and the results presented align with a mouses weekly intake rather than daily.

Round 2

Reviewer 2 Report

Authors have made considerable changes so as to satisfy the points raised by the reviewers.